# Is It Safe to Combine a Fundoplication to Sleeve Gastrectomy? Review of Literature

**DOI:** 10.3390/medicina57040392

**Published:** 2021-04-18

**Authors:** Sergio Carandina, Viola Zulian, Anamaria Nedelcu, Marc Danan, Ramon Vilallonga, David Nocca, Marius Nedelcu

**Affiliations:** 1ELSAN, Clinique Saint Michel, Centre Chirurgical de l’Obésité (CCO), 83100 Toulon, France; sergio.carandina@gmail.com (S.C.); zulian.viola@gmail.com (V.Z.); anamaria.andreica@gmail.com (A.N.); mdanan@wanadoo.fr (M.D.); 2Clinica Madonna della Salute, Department of Digestive and Bariatric Surgery, 45014 Porto Viro, Italy; 3Endocrine, Metabolic and Bariatric Unit, General Surgery Department, Hospital Vall d’Hebron, 08023 Barcelona, Spain; vilallongapuy@gmail.com; 4Faculty of Medicine, Universitat Autònoma de Barcelona, 08023 Barcelona, Spain; 5CHU de Montpellier, 34080 Montpellier, France; d-nocca@chu-montpellier.fr; 6University Montpellier 1, 34080 Montpellier, France; 7ELSAN, Clinique Bouchard, 13000 Marseille, France

**Keywords:** sleeve, GERD, fundoplication, Nissen, Rossetti, Dor, complications

## Abstract

*Background and Objectives:* The rising numbers of laparoscopic sleeve gastrectomy (LSG) procedures now being performed worldwide will likely be followed by an increasing number of patients experiencing gastro-esophageal reflux disease (GERD). The purpose of the current review was to analyze in terms of safety different techniques of fundoplication used to treat GERD associated with LSG. *Methods*: An online search was performed in PubMed/MEDLINE in December 2020 to identify articles reporting LSG and fundoplication. The following term combination was used: (sleeve, fundoplication), (sleeve, Nissen), (sleeve, Rossetti), (sleeve, Toupet) and (sleeve, Dor). The extracted information included details of the methods (e.g., retrospective case series), demographic characteristics (e.g., age, gender), clinical characteristics, number of patients, rate of conversion, and postoperative outcomes. *Results*: A total of 154 studies were identified and after an assessment of title according to our exclusion criteria, 116 articles were removed. Of the 38 studies analyzed for full content review, a total of seven primary studies (487 patients) were identified with all inclusion criteria. Analyzing the different types of fundoplication used, we have identified: 236 cases of Nissen-Sleeve, 220 cases with modified Rossetti fundoplication, 31 cases of Dor fundoplication, and no case of Toupet fundoplication. The overall postoperative complication rate was 9.4%, with the most common reported complication being gastric perforation, 15 cases—3.1%. The second most common complication was bleeding identified in nine cases (1.8%) followed by gastric stenosis in six cases (1.2%). The mortality was nil. *Conclusions*: Different types of fundoplication associated with LSG appear to be a safe surgical technique with an acceptable early postoperative complication rate. Any type of fundoplication associated with LSG to decrease GERD should be evaluated cautiously while prospective clinical randomized trials are needed.

## 1. Introduction

Over the last 10 years, laparoscopic sleeve gastrectomy (LSG) has become the most frequently performed bariatric procedure [1,2,3]. Its success among the other bariatric procedures can be explained by several factors. First, LSG offers similar results comparing to laparoscopic Roux-en-Y gastric bypass (LRYGB) [4] between the effectiveness in treating obesity and related comorbidities in the short- and long-term with good quality of life. Moreover, LSG is considered less technically demanding when compared to other malabsorptive procedures. Unfortunately, as with all other bariatric surgery procedures, sleeve gastrectomy is not exempt from long-term complications. The main long-term side effect of LSG seems to be gastro-esophageal reflux disease (GERD). Following LSG, the presence of GERD symptomatology was reported in up to 20–60% of patients in recent studies [5,6,7,8,9]. For these reasons, in the case of patients presenting with GERD before bariatric surgery, the current recommendation is to propose LRYGB as a primary operation. These recommendations are even stronger when preoperative reflux is associated with a hiatal hernia. Unfortunately, even patients undergoing LRYGB are not completely exempt from the risk of developing postoperative GERD. Additionally, more than 35% of the patients who underwent RYGB had at least one complication within the 10-year follow-up period [10].

Considering all these findings, several bariatric teams proposed a modification to the usual surgical technique of LSG by adding a different fundoplication—Table 1 [11,12,13,14,15,16,17,18]. The intents of these techniques were to minimize the rate of postoperative GERD, to protect the staple line of the angle of His, and finally, to provide a safe and effective alternative for patients with contraindication to LRYGB because of GERD. Even if the initial results are encouraging, performing a valve around the gastric tube remains a sensible technical point that can turn a relatively simple intervention like LSG into a more complex procedure. The aim of this current review was to evaluate different techniques of fundoplication described in the literature associated with LSG to decrease GERD in terms of safety for these investigational procedures.

## 2. Materials and Methods

### 2.1. Search Strategy and Study Selection

An online search was performed in PubMed/MEDLINE in December 2020 to identify articles reporting different techniques of fundoplication associated with LSG. Preferred Reporting Items for Systematic Reviews and Meta-Analyses guidelines were followed for selecting the eligible studies. The following term combination was used: (sleeve, fundoplication), (sleeve, Nissen), (sleeve, Rossetti), (sleeve, Toupet) and (sleeve, Dor). The reference list of the retrieved articles was also manually checked for relevant papers. The conference abstracts considered “gray literature”, were not analyzed. No publication date or language limit was considered for our search strategy.

A total of 154 records were identified by the initial search of the PubMed/MEDLINE databases. Of these, 116 papers were excluded after screening by title and abstract. Studies of any design that involved the results of any technique associating the fundoplication with LSG from 2009 to 2021 were considered. Our initial analysis included a prescreen to identify the clearly irrelevant reports by title, abstract, and keywords of the publication. Two other independent reviewers (D.N. and M.N.) then assessed the studies for relevance, inclusion, and methodological quality. The studies were classified as relevant (meeting all specified inclusion criteria), possibly relevant (meeting some but not all inclusion criteria), and rejected (not relevant to our review). Two reviewers (S.C. and M.D.) independently reviewed the full-text versions of all studies classified as relevant or possibly relevant. Furthermore, the other authors (D.N. and M.N.) have completely analyzed the full text of all relevant studies included in the current final review. Any disagreements were resolved by repeat extraction.

### 2.2. Data Extraction and Management

The extracted information included details of the methods (e.g., retrospective case series), demographic characteristics (e.g., age, gender), clinical characteristics, number of patients, rate of conversion, and postoperative outcomes. All repetitive information was excluded.

### 2.3. Statistical Analysis

We performed an analysis of the data from the included studies. Descriptive statistics (simple counts and mean values) were used to report the study, patient, and treatment-level data. The number of patients enrolled was used in the calculation of the study and patient demographic characteristics. Efficacy outcomes of interest were synthesized. Because of the high heterogeneity among the studies and the complete lack of randomized controlled trials, a meta-analysis was not deemed appropriate.

## 3. Results

### 3.1. Search Results

A total of 154 studies were identified using our search criteria for screening. After an assessment of titles according to our exclusion criteria, 116 articles were removed, and 38 studies remained for full content review. 14 studies were not concerning the topic of the current review. One study [18] was not included in our final analysis despite the extremely interesting and detailed information about the GERD because the author did not report information about the complications following anterior fundoplication associated with LSG.

Of the remaining 23 studies: Seven were excluded for repetitive information, consensus conference, or different types of letters to editors [19,20,21,22,23,24,25], six for their design of case reports [26,27,28,29,30,31], and three for video reports or technical descriptions [32,33,34]. Consequently, in the final analysis of complications, seven primary retrospective studies meeting the inclusion criteria were included.

### 3.2. Included Studies

A total of 487 patients were assessed in the seven studies, and the number of patients included in the different manuscripts ranged from 4 to 220 patients. Different types of fundoplication are described in the literature associated with LSG. Analyzing this criterion, we have identified: 236 cases of Nissen-Sleeve, 220 cases with modified Rossetti fundoplication, 31 cases of Dor fundoplication, and no case of Toupet fundoplication.

There was no mortality in the patient population and the incidence of overall postoperative complication rate was 9.4%, including a rate of 3.1% for minor complications. We have identified a number of 46 patients with complications following different types of fundoplication associated with LSG. The most commonly reported complication was the gastric perforation, 15 cases—3.1%. This was clearly defined as not related to the staple line and additional leak of the staple line was described only in three patients (0.6%). The second most common complication was bleeding identified in nine cases (1.8%), followed by gastric stenosis in six cases (1.2%). All the complications are summarized in Figure 1.

Considering different types of fundoplication we have identified the following rates of complications:-5.7% for Collis Nissen, seven cases out of 122 patients reported by a single team (11).-11.2% for Nissen fundoplication, 13 cases out of 116 patients, procedures performed by four different teams (12, 13, 15, 17, 26, 27)-3.2% for Dor fundoplication, one case out of 31 patients reported by a single team (14).-12.3% for Nissen Rossetti, 27 cases out of 220 patients reported by Olmi et al. by a single team (16).

The reintervention rate was variable from 3% to 8% of the reported experiences and only one conversion to open surgery was reported for bleeding.

## 4. Discussion

The association between obesity and GERD is very well-known. Compared to the general population, in obese patients, GERD symptoms are present in more than 50% of patients, and more than 70% of patients present evidence of reflux disease at PH-metry [35,36]. The pathophysiology of GERD in obese patients is notably multifactorial and it includes an altered gastro-esophageal gradient pressure due to intragastric and intra-abdominal adiposity compression, frequently associated with a hiatal hernia. The use of LSG treatment in patients with preoperative GERD remains controversial. While in some studies, LSG combined with hiatal hernia repair showed good results in controlling GERD, in other recent studies, LSG seems to worsen preexisting GERD and increase the risk of de novo GERD and Barret’s esophagus. The real pathogenesis of GERD after LSG is not completely clear. LSG induces an alteration in the angle of His with consequent hypotony of the lower esophageal sphincter after the division of muscular sling fibers, a decrease in gastric volume that for the Laplace’s law results in an increase in intragastric pressure, a decrease in ghrelin secretion that causes a gastric dysmotility. For all these reasons, a new bariatric technique able to better control GERD in patients with obesity was described in the literature as the association of different types of fundoplication with LSG. This can offer the possibility to perform LSG in patients with preexisting GERD and to decrease the onset of postoperative GERD and Barret’s esophagus in bariatric patients. The downside is the increase in the technical difficulty of an intervention that owes its widespread worldwide diffusion also to its simplicity of execution. Inevitably, adding to the usual operating time of the sleeve, the creation of fundoplication and, above all, the correct section of the stomach in order to preserve its vascularization, increases the difficulty of the intervention and consequently, increases the number of cases necessary to achieve the same proficiency as for LSG. This fact could theoretically increase the risk of postoperative complications. In reality, the creation of the valve, wrapping the region of the cardia allows protecting the upper part of the staple line and the esophago-gastric junction. This region represents an anatomical area of weakness that is more sensitive to any increase in intragastric pressure. For this reason, almost all leaks after LSG originate from this location just below the gasto-esophageal junction. The valve could avoid this potential complication.

Our current review confirmed this hypothesis as the leak from the staple line has a lower frequency, described only in three patients (0.6%) [14,15,17] compared to perforation of the valve described in 15 cases (3.1%) [13,16]. In these cases, the authors recommended laparoscopic revision with resection of the gastric valve, perigastric abscess drainage, and conversion to a standard LSG in the majority of cases. Different manuscripts reported single case complications [21,26,31]. Ben Amor et al., from a cohort study of 70 patients, reported in another manuscript [21] one patient with a fistula at the level of the gastric longitudinal staple line that was successively converted in LRYGB. Similarly, Chouillard et al. [26] presented a clinical case of a patient operated by N-sleeve complicated by two leaks, one anterior at the level of the transection line of the sleeved stomach and the second posterior at the level of the valve. For this case, the patient was converted in LRYGB as well. Since the technique is recent, the literature data are insufficient to confirm whether the creation of the valve has a protective role at the level of the EG junction. The few cases presented seem to confirm this statement, but studies with a higher number of cases are mandatory.

On the other hand, transdiaphragmatic wrap migration, wrap ischemia with perforation, and severe postoperative dysphagia were also described after Nissen’s fundoplication. As it was pointed out from the beginning, there are some technical details that are of paramount importance in valve creation and resection. In the Nissen-Sleeve technique, the vascular supply of the valve comes only from the left gastric artery, so during resection, it is really important to avoid any rotation by applying symmetrical traction through the anterior and posterior wrap wall. In our experience, this is the key point of this procedure. Excessive and asymmetrical traction could generate a valve rotation with vascular damage, whereas insufficient traction could leave a redundant posterior valve wall with consequent food stasis. When a postoperative wrap perforation is diagnosed, as showed by Skalli [31], it can be easily managed by laparoscopic wrap resection that allows a prompt recovery for the patient.

The Roux-en-Y Gastric Bypass is considered the gold standard for GERD, especially following LSG, but recently, new procedures, like single anastomosis sleeve ileal bypass, could be considered a solution. Future prospective clinical randomized trials are needed to validate the benefit of this new procedure especially for the rapid gastric emptying.

## 5. Conclusions

The initial experience with different types of fundoplication associated with LSG appears to be a safe surgical technique with an acceptable early postoperative complication rate. This technique could potentially decrease the risk of leak at the level of the angle of His but on the other hand, it is more technically demanding than the standard LSG, with the risk of increasing the overall rate of complications, especially during the learning curve period. The current data are absolutely insufficient to establish whether one type of fundoplication should be preferable to the others. Any type of fundoplication associated with LSG to decrease GERD should be evaluated cautiously while prospective clinical randomized trials are needed.

## Figures and Tables

**Figure 1 medicina-57-00392-f001:**
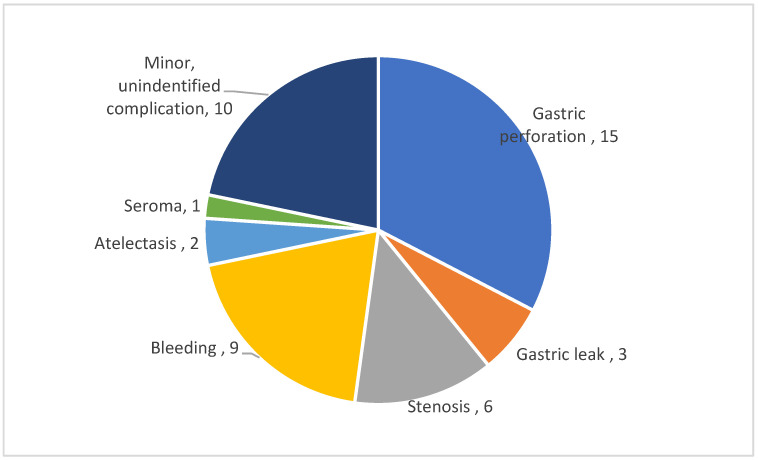
Distribution of identified complications.

**Table 1 medicina-57-00392-t001:** Literature review for fundoplication and sleeve.

Authors	Year	Journal	Number of Cases	Type of Fundoplication	Complication Rate	Type of Complications	Readmission/ Reintervention Rate	Additional Comments
Da Silva et al. [11]	2015	Obesity Surgery	122	Collis Nissen	5.7% −7 cases	Stenosis 4 (3.3%) Atelectasisa 2 (1.6%) GI bleeding 1 (0.8%)	3.3%	Completely different technique with no resection of the gastric fundus
Lepage et al. [12]	2015	World J Surg	4	Nissen	0%	none	N/A	Greatly improvement in gastric emptying
Nocca et al. [13]	2016	SOARD	25	Nissen	20% (2 major + 3 minor)	One bleeding One valve perforation Three minor complications	8%	.
Moon et al. [14]	2016	SOARD	31	Anterior (120°)	3.2%	One leak—conservatory	3.2%	Hiatal hernia was repaired according to intraoperative findings—67% of cases
Lasnibat et al. [15]	2017	ABCD Arq Bras Cir Dig	15	Nissen	13.3% 2 cases	One pneumoperitoneum One seroma	6.6%	six underwent revision surgery, four for weight regain, one for reflux and one mixed.
Olmi et al. [16]	2020	SOARD	220	modified Rossetti	12.3% 27 cases	Gastric perforation 14 cases Bleeding 6 cases	6.4%	Improvement in esophagitis was observed in 63 of 65 (96.92%) patients.
Ben Amor et al. [17]	2020	Obesity Surgery	70	Nissen	5.7% 4 cases	Two stenosis One fistula One bleeding	3%	

## Data Availability

Not applicable.

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
