# Peer review of "Is It Safe to Combine a Fundoplication to Sleeve Gastrectomy? Review of Literature"

_medicina, 2021, doi:10.3390/medicina57040392_

Round 1

Reviewer 1 Report

Overall, the authors did a good job describing the major downfall of a laparoscopic sleeve gastrectomy being reflux disease. They offer insight into a review of the literature as offering a concomitant antireflux procedure to help mitigate this. They did a nice job identifying and summarizing the data, but more importantly highlight that we still cannot make definitive conclusions on the current data. They also mention the implications/complications associated with a concomitant wrap with a sleeve gastrectomy. My only criticism is in the results section there are some grammatical errors/wording that should be worked out. 

Author Response

We thank the reviewers for their fair and very constructive feedback. We have done the appropriate modifications according to our experience and convictions. We are convinced that by the modifications done to the manuscript according to your suggestions we have highly improved the quality of our paper.

Review #1

Overall, the authors did a good job describing the major downfall of a laparoscopic sleeve gastrectomy being reflux disease. They offer insight into a review of the literature as offering a concomitant antireflux procedure to help mitigate this. They did a nice job identifying and summarizing the data, but more importantly highlight that we still cannot make definitive conclusions on the current data. They also mention the implications/complications associated with a concomitant wrap with a sleeve gastrectomy. My only criticism is in the results section there are some grammatical errors/wording that should be worked out.

Thank you for the positive comment and evaluation. The purpose of current manuscript was to put together all the manuscripts regarding different types of fundoplication around the gastric tube performed during sleeve in order to prevent GERD. The current data are too scarce to get valid conclusions, but as you have mentioned, the manuscript identify and summarize all the available data. The results section was revised, and we hope now you will find it satisfactory for the standards of “Medicina” and suitable for publication.

Reviewer 2 Report

Review of paper titled ‘Is it safe to combine a fundoplication to sleeve gastrectomy? Review of literature.’’

Thank you for the invitation to review this article.

Abstract: Nicely written.

Introduction: Well written

Method

We know that majority of the single complications are published as case reports. Also majority of the journals now don’t accept case reports. So some are reported as letter to editor or as videos. They need to be included in the final result for the true representation of the complications:

Eg:

Ref 21 – Chronic fistula post Nissen SG.

Ref 26 – Complex leak after the ‘’Nissen’’ variant of Sleeve gastrectomy

Ref 27 – Ischaemia of the wrap after Nissen SG

Ref 31 – Wrap necrosis after Nissen SG.

These cases need to be included in the table 1 to give true representation.

Result:

Would suggest that the authors separate Collis Nissen (122 cases) and Nissen- Sleeve (114) as they both are technically different. There is difference in the position of the proximal staple line.

They need to mention cumulative results of different procedures separately to give an idea which procedure (Collis, Nissen, anterior or Rossetti) is safer than the other.

Eg Collis Nissen – 122 cases, 5.7% complication rate.

Eg Nissen-SG, Ref 12, 13, 15 and 17 cumulative shows 114 cases. They show total complication rate of 39%. This is very high. Is that safe? Also the authors have not included the ref 21, 26, 27, and 31 in this calculation which adds to the complications. The readmission/reintervention rate is also around 17.6%. This is high compared to other fundoplication.

The perforation and leak rate is of total 3.7% (3.1% + 0.6%). Is this not high for primary bariatric procedures?

The authors need to mention the duration of follow up and % of follow up of patients in table 1.

Discussion:

Majority of these study report only 1 year follow up. This short term follow up need to be highlighted in the study. They need to mention that we do not know what happens to GERD in long term. We know that in case of SG the GERD rate is high in long term. Also in pure fundoplications the recurrence rate of GERD in long term are higher.

Conclusion:

Based on the limited data available Collis nissen (n=122) seem to be safe compared to anterior wrap (n=only 31 patients) compared to modified rossetti (n=220). Nissen-sleeve (n=114) has the highest complication rate compared to others.

The literature is very sparse to conclude safety. The follow up is also very short.

Decision:

My humble opinion is that the article needs more details. The article is interesting but needs the above points addressed please. 

Author Response

Review #2

Review of paper titled ‘Is it safe to combine a fundoplication to sleeve gastrectomy? Review of literature.’’Thank you for the invitation to review this article.

Abstract: Nicely written.

Introduction: Well written

Method

We know that majority of the single complications are published as case reports. Also majority of the journals now don’t accept case reports. So some are reported as letter to editor or as videos. They need to be included in the final result for the true representation of the complications:

Eg: Ref 21 – Chronic fistula post Nissen SG.

Ref 26 – Complex leak after the ‘’Nissen’’ variant of Sleeve gastrectomy

Ref 27 – Ischaemia of the wrap after Nissen SG

Ref 31 – Wrap necrosis after Nissen SG.

These cases need to be included in the table 1 to give true representation.

We completely agree with you that also the case reports are part of the entire experience with fundus plication around the gastric tube. According to your recommendation, we have included two references (26 and 27) in a revised form of the Table 1. The refereces 21 and 31 were not included in tha table as they are redundant and these case reports were already included in their entire experience – see references 13 and 17.

Result:

Would suggest that the authors separate Collis Nissen (122 cases) and Nissen- Sleeve (114) as they both are technically different. There is difference in the position of the proximal staple line.

They need to mention cumulative results of different procedures separately to give an idea which procedure (Collis, Nissen, anterior or Rossetti) is safer than the other.

Eg Collis Nissen – 122 cases, 5.7% complication rate.

Eg Nissen-SG, Ref 12, 13, 15 and 17 cumulative shows 114 cases. They show total complication rate of 39%. This is very high. Is that safe? Also the authors have not included the ref 21, 26, 27, and 31 in this calculation which adds to the complications. The readmission/reintervention rate is also around 17.6%. This is high compared to other fundoplication.The perforation and leak rate is of total 3.7% (3.1% + 0.6%). Is this not high for primary bariatric procedures?

Thank you very much for this suggestion. Accordingly, the following paragraph was added to the results: “Considering different types of fundoplication we have identified the following rates of complications:

  • 7 % for Collis Nissen, 7 cases out of 122 patients reported by a single team (11).
  • 2 % for Nissen fundoplication, 13 cases out of 116 patients, procedures performed by 4 different teams (12, 13, 15, 17, 26, 27)
  • 2 % for Dor fundoplication, one case out of 31 patients reported by a single team (14).
  • 3 % for Nissen Rossetti, 27 cases out of 220 patients reported by Olmi et al. by single team (16).”

We cannot explain your percentage of 39 % !!! Our review found 13 cases of complications out of 116 patients, which represent 11.2 %. Even so, the percentage could be considered extremely high for a primary bariatric procedure but could also be explained by the learning curve process and the fact that Nissen fundoplication is a complete (360 °) wrap which is more prone to complications.

Discussion:

Majority of these study report only 1 year follow up. This short term follow up need to be highlighted in the study. They need to mention that we do not know what happens to GERD in long term. We know that in case of SG the GERD rate is high in long term. Also in pure fundoplications the recurrence rate of GERD in long term are higher.

We cannot agree more with your comment for the need of long term follow up. Even so the purpose of the current review was more focused on safety, having the principal endpoint the rate of complication. In the future, these procedures should be also evaluated in term of efficacy for GERD as you have emphasized.

Conclusion:

Based on the limited data available Collis nissen (n=122) seem to be safe compared to anterior wrap (n=only 31 patients) compared to modified rossetti (n=220). Nissen-sleeve (n=114) has the highest complication rate compared to others.

The literature is very sparse to conclude safety. The follow up is also very short.

According to your recommendations, the conclusions were revised as follow: “The initial experience with different types of fundoplication associated to LSG appears to be a safe surgical technique with an acceptable early postoperative complication rate. This technique could potentially decrease the risk of leak at the level of the angle of His but on the other hand is more technical demanding than the standard LSG, with the risk of increasing the overall rate of complications, especially during the learning curve period. The current data are absolutely insufficient to establish whether one type of fundoplication should be preferrable to the others. Any type of fundoplication associated to LSG to decrease GERD should be evaluated cautiously while prospective clinical randomized trials are needed.”

Decision:

My humble opinion is that the article needs more details. The article is interesting but needs the above points addressed please. 

We are convinced that by the modifications done to the manuscript according to your suggestions we have highly improved the quality of our paper. We hope now you will find it satisfactory for the standards of “Medicina” and suitable for publication.

Reviewer 3 Report

Overall, this is a well written review of the literature regarding the safety of fundoplication in combination with laparoscopic sleeve gastrectomy. I only have a few minor comments:

  1. A surprisingly high number of initially identified papers was excluded from the study (116 out of 154 = 75%). This puts the search strategy into questions and may indicate selection bias. Please comment on this point in the discussion.
  2. Although the paper focuses on safety of fundoplication in combination with LSG, different approaches to address the problem of post LSG reflux should be discussed at least briefly; e.g. SASI bypass which accelerates gastric emptying and relieves the high pressure of LSG.

Author Response

Review #3

Overall, this is a well written review of the literature regarding the safety of fundoplication in combination with laparoscopic sleeve gastrectomy. I only have a few minor comments:

Thank you for the positive comment and evaluation. According to all reviewers’ comments, we have done the appropriate modifications according to our experience and convictions. We are convinced that by the modifications done to the manuscript according to your suggestions we have highly improved the quality of our paper.

  1. A surprisingly high number of initially identified papers was excluded from the study (116 out of 154 = 75%). This puts the search strategy into questions and may indicate selection bias. Please comment on this point in the discussion.

We completely agree with you with this common problem, but in every review of the literature following the first search, a lot of titles are not significant for the purpose of the review.  In our case, with different terms of fundoplication many titles were selected with no relevance for concomitant fundoplication and sleeve.

  1. Although the paper focuses on safety of fundoplication in combination with LSG, different approaches to address the problem of post LSG reflux should be discussed at least briefly; e.g. SASI bypass which accelerates gastric emptying and relieves the high pressure of LSG.

Thank you very much for this comment. We have also some past positive experiences with SASI bypass both for weight regain following sleeve or for important metabolic syndrome. Unfortunately, at the current time this procedure was forbidden in our country. According to your recommendation, the following paragraph was added to the discussion:

“The Roux-en-Y Gastric Bypass is considered the gold standard for GERD especially following LSG, but recently new procedure like single anastomosis sleeve ileal bypass could be considered a solution Future prospective clinical randomized trials are needed to validate the benefit of this new procedure especially for the rapid gastric emptying.”

Round 2

Reviewer 2 Report

Satisfactory corrections. Thank you.